# Locking loop movement in the ubiquinone pocket of complex I disengages the proton pumps

Alfredo Cabrera-Orefice[1,2], Etienne Galemou Yoga[3,4], Christophe Wirth [5], Karin Siegmund[3,4], Klaus Zwicker [6], Sergio Guerrero-Castillo[1], Volker Zickermann [2,3,4], Carola Hunte [5] & Ulrich Brandt [1,2]

Complex I (proton-pumping NADH:ubiquinone oxidoreductase) is the largest enzyme of the mitochondrial respiratory chain and a significant source of reactive oxygen species (ROS). We hypothesized that during energy conversion by complex I, electron transfer onto ubiquinone triggers the concerted rearrangement of three protein loops of subunits ND1, ND3, and 49-kDa thereby generating the power-stoke driving proton pumping. Here we show that fixing loop TMH1-2[ND3] to the nearby subunit PSST via a disulfide bridge introduced by site-directed mutagenesis reversibly disengages proton pumping without impairing ubiquinone reduction, inhibitor binding or the Active/Deactive transition. The X-ray structure of mutant complex I indicates that the disulfide bridge immobilizes but does not displace the tip of loop TMH1-2[ND3]. We conclude that movement of loop TMH1-2[ND3] located at the ubiquinone-binding pocket is required to drive proton pumping corroborating one of the central predictions of our model for the mechanism of energy conversion by complex I proposed earlier.

[1] Radboud Institute for Molecular Life Sciences, Department of Pediatrics, Radboud University Medical Center, Geert Grooteplein-Zuid 10, 6525, GA Nijmegen, The Netherlands. [2] Cluster of Excellence Macromolecular Complexes, Goethe-University, Max von Laue Str. 9, 60438 Frankfurt am Main, Germany. [3] Institute of Biochemistry II, Medical School, Goethe University, Max von Laue Str. 9, 60438 Frankfurt am Main, Germany. [4] Centre for Biomolecular Magnetic Resonance, Institute for Biophysical Chemistry, Goethe University, Max von Laue Str. 9, 60438 Frankfurt am Main, Germany. [5] Institute for Biochemistry and Molecular Biology, ZBMZ, Faculty of Medicine, BIOSS Centre for Biological Signalling Studies, University of Freiburg, Stefan-Meier-Str. 17, 79104 Freiburg im Breisgau, Germany. [6] Institute of Biochemistry I, Medical School, Goethe University, Theodor-Stern-Kai 7, 60590 Frankfurt am Main, Germany. Correspondence and requests for materials should be addressed to U.B. (email: ulrich.brandt@radboudumc.nl)

Respiratory complex I feeds redox equivalents into oxidative phosphorylation. By linking electron transfer from NADH to ubiquinone with the translocation of four protons across the inner mitochondrial membrane, it generates a major portion of the proton motive force driving aerobic ATP synthesis and the transport of metabolites. Complex I is also a key player in signaling and pathologic mechanisms associated with reactive oxygen species (ROS) in humans[1]. Yet, the molecular mechanism and regulation of energy conversion and ROS production by complex I remain poorly understood. Recent structures of complex I from different species obtained by X-ray crystallography[2,3] and cryo-electron microscopy[4–7] showed that FMN and a chain of seven iron–sulfur clusters reside in its peripheral arm (N/Q modules) and connect the NADH oxidation site to the ubiquinone reduction site. As had been proposed earlier[8,9], the latter is located significantly above the membrane surface and remote from the four putative pump sites distributed over the long membrane arm (P module) of complex I[2] (Supplementary Fig. 1a). It has been proposed that energy transmission between the pump sites occurs via electrostatic and conformational transmission along an axis of protonable residues at the center of the membrane arm[10] and this concept is supported by recent molecular modeling studies[11,12]. It is still a matter of debate, how the energy released during ubiquinone reduction is converted into the power stroke sent along the P-module to drive the proton pumps of complex I.

If substrates are added to complex I from vertebrates and fungi that has been idling for some time at physiological temperatures, activity is very low initially, but then returns to maximal levels[13,14]. The physiological role of this so called Active/Deactive (A/D) transition of mitochondrial complex I remains unclear, but locking the enzyme in the D-form prevents cardiac reperfusion injury in vivo[15]. Interconversion from D-form to A-form can be blocked by divalent cations ($Me^{2+}$), alkaline pH or by any kind of covalent modification of a single cysteine residue[16] that is only accessible in the D-form[17]. This cysteine resides in the hydrophilic TMH1-2 loop of the mitochondrially-encoded subunit ND3[18] putatively involved in catalysis[2] and is thus also ideally positioned to control the A/D transition of complex I.

Based on insights from the X-ray structure of *Yarrowia lipolytica* complex I, we hypothesized[2] that during turnover negatively charged ubiquinone intermediates trigger the concerted movement of three protein loops to generate the power stroke driving the proton pumps, in line with the principles of the earlier proposed two-state stabilization change mechanism[19] (Supplementary Fig. 1b). These loops are (a) the $\beta_1$–$\beta_2$ loop of 49-kDa subunit involved in binding the ubiquinone headgroup; (b) the TMH1-2 loop of subunit ND3 and (c) the highly acidic TMH5-6 loop of subunit ND1. In the cryo-EM[4–7] structures of mammalian complex I[3,4] these loops exhibited pronounced structural flexibility lending support to our mechanistic hypothesis[2]. In fact, in the recently obtained cryo-EM structures of the deactive enzyme several domains near the ubiquinone-binding pocket were disordered, which included major parts of the respective loops in the 49-kDa and ND3 subunits[5,6]. In contrast, in the X-ray structure of the deactive form of *Y. lipolytica* complex I the corresponding loops showed continuous density except for loop TMH1-2$^{ND3}$, for which only the protein backbone of the stretch in immediate contact with the ubiquinone-binding pocket could be modelled[2]. The varying degree by which these loops are disordered in the deactive form of the enzyme may reflect the differences in the stability and energetics of the A/D conversion between mammalian and fungal complex I reported earlier[14]. Moreover, molecular modeling studies point toward functional importance of structural rearrangements involving the same loop domains[11,12].

Here we show that blocking the movement of loop TMH1-2$^{ND3}$ by tethering it to subunit PSST by an engineered disulfide bridge reversibly disengages the proton pumps of mitochondrial complex I, without affecting kinetic parameters of ubiquinone reduction, A/D transition or proton permeability of the membrane domain. We conclude that movement of loop TMH1-2$^{ND3}$ is required to drive the proton pumps of complex I.

## Results

**Introducing a disulfide bridge to fix loop TMH1-2$^{ND3}$.** We reasoned that, if movement of the three loops lining the ubiquinone-binding cavity was indeed required for the mechanism and regulation of mitochondrial complex I, locking one of them should interfere with ubiquinone reduction, proton pumping and/or A/D transition. To fix loop TMH1-2$^{ND3}$ by introducing an artificial disulfide bridge, we took advantage of the available structures of complex I and identified suitable positions in subunits PSST and 49-kDa for introducing a second cysteine near C40 of loop TMH1-2$^{ND3}$. Making use of the genetic accessibility of the yeast *Y. lipolytica*, we replaced Q133 in subunit PSST (NUKM in *Y. lipolytica*) by a cysteine using site-directed mutagenesis. When we treated complex I purified from strain Q133C$^{PSST}$ with Ellman's reagent (5,5′-dithiobis-2-nitrobenzoic acid; DTNB)[20], which promotes the formation of disulfide bridges from adjacent cysteine-thiols, an additional band occurred in non-reducing SDS-PAGE migrating between subunits ND1 and 30-kDa (Fig. 1a). The band was not observed in complex I from the parental strain or after reduction with dithiothreitol (DTT). The presence of a disulfide cross-link between subunits ND3 and PSST was confirmed by LC-ESI tandem mass spectrometry (MS) (Fig. 1b) and also by separating the two proteins in a second dimension SDS-PAGE at reducing conditions (Fig. 1c). Furthermore, non-crosslinked subunits PSST and ND3 were hardly detectable by MS in DTNB-treated complex I from strain Q133C$^{PSST}$ (Supplementary Fig. 2) demonstrating almost complete formation of the inter-subunit disulfide bridge.

**No effect of Q133C$^{PSST}$ on assembly and enzyme kinetics.** Blue-native (BN) PAGE of mitochondrial membranes from strain Q133C$^{PSST}$ revealed that assembly and content of complex I were not affected by the mutation (Supplementary Fig. 3a). Confirming normal content of complex I and indicating a fully functional NADH oxidation site, deamino-NADH:hexaammineruthenium (dNADH:HAR) oxidoreductase activity was unchanged (Supplementary Fig. 3b) and inhibitor-sensitive dNADH:decylubiquinone (dNADH:DBQ) oxidoreductase appeared only 10–15% lower barely reaching statistical significance (Table 1; Supplementary Fig. 3c). With purified complex I, treatment with DTNB reduced all activities by ~10%, and this minor effect did not correlate with cross-link formation in mutant Q133C$^{PSST}$ (Tables 1, 2). Moreover, the minimal effect on maximal turnover also excluded significant interference by stable modifications of the cysteines by the reagent under the conditions used. Major inactivation of complex I was however observed, when much longer incubation times (>15 min) than those applied in the experiments described here (5 min) were applied. No significant differences in $V_{max}$ and $K_M$ values for decylubiquinone were observed between parental and mutant complex I (Table 1, Supplementary Fig. 4a, b). The sensitivities to the specific complex I inhibitors *n*-decyl-quinazoline-amine (DQA) and rotenone were the same in parental and mutant complex I (Table 1; Supplementary Fig. 4c-f). No effects of mutation Q133C$^{PSST}$ on the immediate electron donor for ubiquinone, iron–sulfur cluster N2, and other iron–sulfur clusters were detected by continuous-wave electron paramagnetic resonance (cw-EPR) spectroscopy of

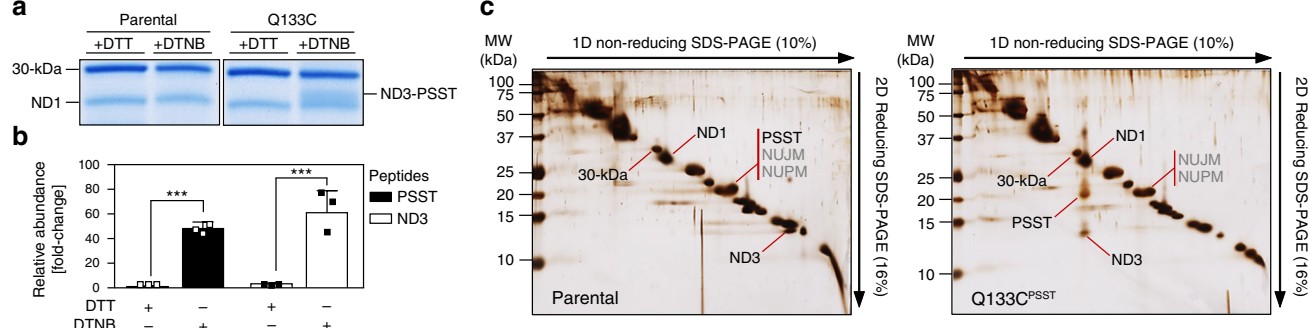

**Fig. 1** A disulfide cross-link is formed between subunits ND3 and PSST in mutant Q133C$^{PSST}$. Purified complex I from parental and mutant strains were incubated with either 5 mM dithiothreitol (DTT) or 0.1 mM 5,5'-dithiobis-2-nitrobenzoic acid (DTNB) for 5 min and then separated by non-reducing Tricine SDS-PAGE. **a** An additional band was observed in the sample from mutant Q133C$^{PSST}$ treated with DTNB. **b** Mass spectrometric analysis of the corresponding gel slice confirmed the presence of cross-linked subunits PSST and ND3 and allowed label free quantification of their relative abundance as compared to the corresponding gel slice of the DTT treated sample (mean ± s.d.; $n = 3$ technical replicates; ***$p < 0.001$, ANOVA with Bonferroni correction). **c** Subunits PSST and ND3 were also identified by dSDS-PAGE, in which the cross-link was preserved in the first dimension (1D) and then reduced in the second dimension (2D) to separate both proteins at a shifted position in the mutant. Note that the spot containing subunit PSST does not disappear after cross-linking, because it also contains accessory subunits NUJM and NUPM[37]

---

**Table 1 Functional characterization of complex I in mitochondrial membranes**

| Strain | Treatment[a] | HAR activity[b] | $V_{max}$[c] | $K_M$[d] | IC$_{50}$[e] | |
|---|---|---|---|---|---|---|
| | | | | | DQA | Rotenone |
| | | μmol min$^{-1}$ mg$^{-1}$ | μmol min$^{-1}$ mg$^{-1}$ | μM | nM | nM |
| Parental | – | 0.860 ± 0.02 | 0.193 ± 0.02 | n.d.[f] | n.d. | n.d. |
| | DTT | 0.842 ± 0.04 | 0.160 ± 0.02 | 17 ± 3 | 16 ± 3 | 560 ± 55 |
| | DTNB | 0.726 ± 0.06 | 0.139 ± 0.07 | 20 ± 2 | 12 ± 2 | 524 ± 70 |
| Q133C$^{PSST}$ | – | 0.834 ± 0.11 | 0.165 ± 0.01 | n.d. | n.d. | n.d. |
| | DTT | 0.814 ± 0.06 | 0.143 ± 0.02 | 19 ± 3 | 13 ± 3 | 598 ± 65 |
| | DTNB | 0.688 ± 0.04 | 0.136 ± 0.05 | 21 ± 3 | 17 ± 3 | 593 ± 82 |

[a]Mitochondrial membranes were incubated with 5 mM DTT or 0.1 mM DTNB for 5 min before assay. Three independent batches of membranes were analyzed. Data are given as mean ± s.d.
[b]dNADH:HAR oxidoreductase activity
[c]To account for variations in complex I content in different batches of mitochondrial membranes, dNADH:DBQ oxidoreductase activities were normalized to their specific HAR activities
[d]Apparent $K_M$ for decylubiquinone (DBQ) was determined by fixing the [dNADH] to 100 μM and varying the [DBQ] between 0 and 100 μM
[e]IC$_{50}$ values were determined from curves shown in Supplementary Fig. 4. dNADH:DBQ oxidoreductase activities were measured in the presence of both substrates at 100 μM and varying the inhibitor concentration
[f]Not determined

---

**Table 2 Activities of purified complex I reconstituted into proteoliposomes**

| Strain | Treatment[a] | Complex I activity[b] | | Coupling ratio[c] |
|---|---|---|---|---|
| | | −CCCP | +CCCP | |
| | | μmol min$^{-1}$ mg$^{-1}$ | | |
| Parental | – | 4.6 ± 1.1 | 12.5 ± 1.9 | 2.7 |
| | DTT | 4.9 ± 1.4 | 13.1 ± 2.1 | 2.7 |
| | DTNB | 4.5 ± 1.5 | 11.8 ± 2.4 | 2.6 |
| Q133C$^{PSST}$ | – | 8.4 ± 0.8 | 9.7 ± 0.5 | 1.2 |
| | DTT | 4.8 ± 1.0 | 10.4 ± 0.4 | 2.2 |
| | DTNB | 8.1 ± 1.0 | 9.5 ± 0.4 | 1.1 |

[a]Proteoliposomes were incubated with 5 mM DTT or 0.1 mM DTNB for 5 min before oxidoreductase activity measurements. Three independent batches of proteoliposomes were analyzed. Data are given as mean ± s.d.
[b]DQA-sensitive NADH:DBQ oxidoreductase activities were measured in the absence or presence of 5 μM CCCP. To correct for variations in content and orientation of complex I in the proteoliposomes, values were normalized to their respective NADH:HAR oxidoreductase activities, which were 40 ± 1 μmol min$^{-1}$ mg$^{-1}$ and 35 ± 1 μmol min$^{-1}$ mg$^{-1}$ for parental and mutant Q133C$^{PSST}$ samples, respectively
[c]Coupling ratios were calculated by dividing the NADH:DBQ oxidoreductase activities in the presence of 5 μM CCCP by the activities in its absence

purified complex I (Supplementary Fig. 5). Treatment with DTT and DTNB had some minor effects on the intensity and shape of the EPR signatures, but these were the same with parental and mutant enzyme. Taken together, functional analysis demonstrated that mutation Q133C$^{PSST}$ had no detectable effect on redox centers and the electron transfer activities within the peripheral arm of complex I indicating that the structure of the domains executing these functions were essentially unchanged.

**No effect of Q133C$^{PSST}$ on A/D transition**. To explore whether the A/D transition was affected by blocking the movement of the loop TMH1-2$^{ND3}$ carrying the cysteine only accessible in the D-form, we measured complex I activities in the presence of MgCl$_2$ or $N$-ethylmaleimide (NEM), which react with the D-form only and prevent its transition to the A-form in a reversible or irreversible fashion, respectively[16]. Note that different than the mammalian enzyme[16], Mg$^{2+}$ locks complex I from *Y. lipolytica* very efficiently and stably in the D-form making it a valid tool to assay the D-form (Supplementary Fig. 6). Mg$^{2+}$ prevented reactivation of complex I from both parental and strain Q133C$^{PSST}$ to very similar extents after DTT and DTNB treatment (Fig. 2a). In mutant Q133C$^{PSST}$ treated with DTNB, NEM locked only a small

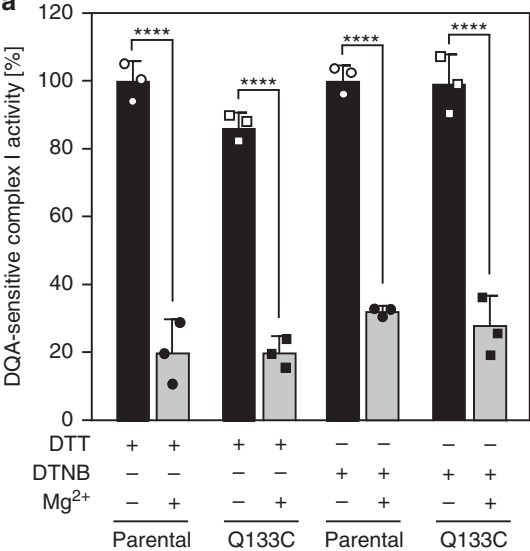

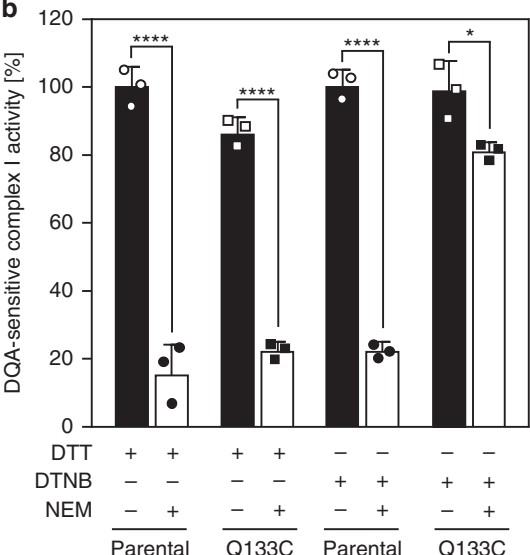

**Fig. 2** Fixing loop TMH1-2$^{ND3}$ does not interfere with complex I activity and A/D transition. Mitochondrial membranes from parental and mutant strains were incubated with 5 mM dithiothreitol (DTT) or 0.1 mM 5,5′-dithiobis-2-nitrobenzoic acid (DTNB) for 5 min at room temperature. Complex I activities were measured in the absence and presence of 5 mM MgCl$_2$ (**a**) or 2 mM *n*-ethylmaleimide (NEM) (**b**). In all cases, 1 mM decylubiquinone (DBQ) was added to the samples before the reaction was started by the addition of DBQ activity buffer containing 110 μM dNADH and supplemented with either 5.5 mM MgCl$_2$ or 2.2 mM NEM. The final concentrations of DBQ and membranes were 100 μM and 50 μg ml$^{-1}$, respectively. *n*-Decyl-quinazoline-amine (DQA)-sensitive dNADH:DBQ oxidoreductase activities in Q133C$^{PSST}$ normalized to their respective controls (parental) are indicated. Activities were measured at pH 8.5 since the inhibition of the D-form by divalent cations is stronger at alkaline pH values, as well as it favors the labeling of the exposed C40$^{ND3}$ by NEM. Data from three independent experiments (mean ± s.d.) are shown. $*p <$ 0.05; $***p <$ 0.001; $****p <$ 0.0001, ANOVA with Bonferroni correction

fraction of complex I in the D-form (Fig. 2b). This was expected, since under these conditions, the cysteine in loop TMH1-2$^{ND3}$ was engaged in the cross-link with subunit PSST and thus was not available to react with the thiol-agent. It follows that formation of

the disulfide bridge attaching loop THM1-2$^{ND3}$ to subunit PSST had no effect on the A/D transition. This is remarkable, since covalently alkylating C40$^{ND3}$ by NEM after breaking the disulfide with DTT still locked complex I from mutant Q133C$^{PSST}$ in the D-form.

**Fixing loop THM1-2$^{ND3}$ reversibly disengages the proton pumps.** To test whether the Q133C$^{PSST}$ mutation affected proton pumping, we reconstituted purified complex I into proteoliposomes and measured pH gradient formation by monitoring the quenching of 9-amino-6-chloro-2-methoxyacridine (ACMA). After addition of DTT, mutant complex I was clearly able to pump protons, although at somewhat reduced efficiency compared to control (Fig. 3a, b). Assuming similar electron transfer rates, this could be either due to a lower pumping stoichiometry or due to slipping of the proton pumping machinery. In contrast, when the disulfide crosslink was formed in the mutant complex I by incubation with DTNB, pH gradient formation was abolished almost completely, while it had no effect at all in the control (Fig. 3c, d). This shows that blocking the movement of the ND3 loop specifically disengaged the proton pumps. Importantly, this block was found to be fully reversible, since releasing the loop by adding DTT during the assay restored proton pumping within seconds (Fig. 3e, f). Declutching of the pumps by the intersubunit crosslink was further corroborated by the finding that the activity of mutant complex I could be increased significantly by carbonylcyanide *m*-chlorophenylhydrazine (CCCP) only when the disulfide bond was broken, while even in the absence of the protonophore rates were already at uncoupled levels after DTNB treatment (Table 2). Notably, untreated liposomes containing complex I from mutant Q133C$^{PSST}$ were also largely uncoupled suggesting the disulfide bridge in most of the enzyme had already formed over time by air oxidation.

**Disengaging the proton pumps does not cause a proton leak.** Roberts and Hirst[21] reported that the deactive form exhibits Na$^+$/H$^+$ antiporter activity for mammalian complex I, although it should be noted that no indications for Na$^+$ transport by complex I were found for the *Y. lipolytica* enzyme[22]. Yet, it seemed conceivable that disconnecting the proton pumps of complex I from the ubiquinone reaction machinery could create a proton leak in the P-module. To assess the proton leak, we followed the decay of a proton gradient across the membrane of complex I proteoliposomes created by the addition of an aliquot of HCl and measured the formation of a proton gradient driven by an electric membrane potential generated by applying a K$^+$ gradient and adding the K$^+$ specific ionophore valinomycin. As shown in Fig. 4, the rate of the non-enzymatic proton leak was identical for proteoliposomes containing complex I from the parental strain and mutant Q133C$^{PSST}$, both under oxidizing and reducing conditions. We concluded that disengaging the proton pumps by tethering loop TMH1-2$^{ND3}$ to subunit PSST did not increase the permeability of the P-module of complex I for protons.

**The disulfide bridge does not displace loop THM1-2$^{ND3}$.** To understand the observed changes at the atomic level, we crystallized complex I and determined the X-ray structure of variant Q133C$^{PSST}$ under conditions favoring formation of the disulfide bond (Supplementary Table 1). Taking advantage of the anomalous signal of sulfur atoms at 5.975 keV[23,24], a strong peak in the anomalous Fourier electron density map indeed indicated presence and location of a disulfide bond (Fig. 5, Supplementary Table 1). The structure was refined at an anisotropic resolution of 4.2 × 4.2 × 3.8 Å. It is overall very similar to the previously

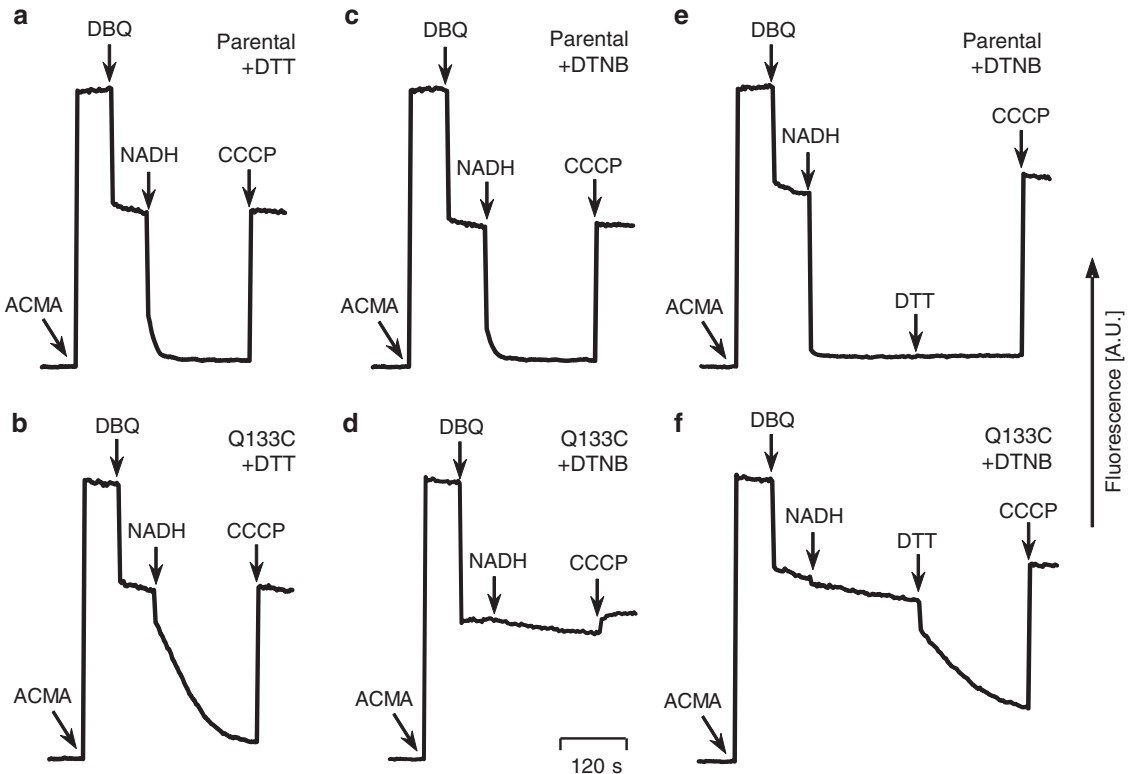

**Fig. 3** Blocking movement of loop TMH1-2[ND3] reversibly disengages the proton pumps. Proton pumping was monitored as 9-amino-6-chloro-2-methoxyacridine (ACMA) fluorescence quench in proteoliposomes containing either parental or mutant complex I treated with 5 mM dithiothreitol (DTT) (**a**, **b**) or 0.1 mM 5,5'-dithiobis-2-nitrobenzoic acid (DTNB) (**c–f**) for 5 min before starting the assay. As indicated, 0.5 μM ACMA, 70 μM decylubiquinone (DBQ). 100 μM NADH and 5 μM carbonylcyanide *m*-chlorophenylhydrazine (CCCP) were added during the assay. Addition of 5 mM DTT to DTNB treated proteoliposomes during the assay had no effect on parental complex I (**e**), but rapidly restored proton pumping of the mutant enzyme (**f**). Representative traces of one out of four datasets obtained with independent batches of proteoliposomes are shown

published structure of *Y. lipolytica* complex I in the deactive form[2]. In the structure of the variant, the central part of the TMH1-2[ND3] loop is ordered (residues 38 to 46). The respective connections to the helices TMH1 and THM2 lack electron density indicating that they are most likely flexible and were thus omitted from the structure. Residue C40 of loop TMH1-2[ND3] and residue Q133C of subunit PSST form a disulfide bond (Fig. 5, Supplementary Fig. 7). This covalent link positions the tip of loop TMH1-2[ND3] close to strand β1[49-kDa]. The tip protrudes into the space between the 49-kDa, the PSST and the ND1 subunits. Notably, the main chain of the TMH1-2[ND3] loop is in a very similar position compared to that in the structure of wild-type complex I (Fig. 5). The side chain positions are not resolved in the latter structure despite the slightly better resolution[2]. In addition, the sulfur anomalous Fourier electron density map for wild-type complex I lacks a peak at the position of C40 of loop TMH1-2[ND3] (Supplementary Table 1). This indicates higher flexibility of side chain and loop, in line with the availability of the cysteine-thiol for chemical modification in the deactive form. Thus, the disulfide bond formation in Q133C[PSST] variant tethers the TMH1-2[ND3] loop in a position very similar to the one it assumes in the deactive form of the wild-type complex.

## Discussion

In summary, our results corroborate one of the central predictions of the mechanistic hypothesis presented earlier[2] that energy transmission from the site of ubiquinone reduction to remotely located proton pumps of complex I requires the movement of

TMH1-2[ND3] as part of coordinated rearrangements of three loops in subunits ND1, 49-kDa, and ND3. At the structural level, the most plausible, although at this point speculative interpretation of our findings would be as follows: Loop TMH1-2[ND3] is required to transmit the conformational changes induced through the reduction of ubiquinone in loop β1–β2[49-kDa] to loop TMH5-6[ND1]. The disulfide bridge in the mutant complex fixes it in a position, which prevents this transmission by keeping the loop from properly interacting with the other loops, without interfering with the other functionalities of the ubiquinone reactive site formed primarily by domains of the 49-kDa and the PSST subunits. Unexpectedly, fixing loop TMH1-2[ND3] at C40 seemed not to interfere with the A/D transition, while any modification of this residue by a thiol agent locks complex I in the inactive D-form. This can be reconciled by assuming that the modified but mobile loop can still engage with loop β1–β2[49-kDa], but that the substituent blocks their functional rearrangements at a critical point thereby preventing catalytic turnover. Our results also seem to suggest that the differential accessibility of loop TMH1-2[ND3] has to be considered mainly as a reporter for the A/D transition. The regulatory transitions of complex I themselves rather seem to primarily involve loop β1–β2[49-kDa] and possibly some of the accessory subunits surrounding it, as suggested by the recent structures of the active and deactive form of mammalian complex I[6].

More detailed data at high resolution ideally representing different reaction intermediates, modeling studies, and functional data will be needed to decipher the conformational changes associated with complex I turnover in detail. Being now able to disconnect selectively and reversibly the proton pumping

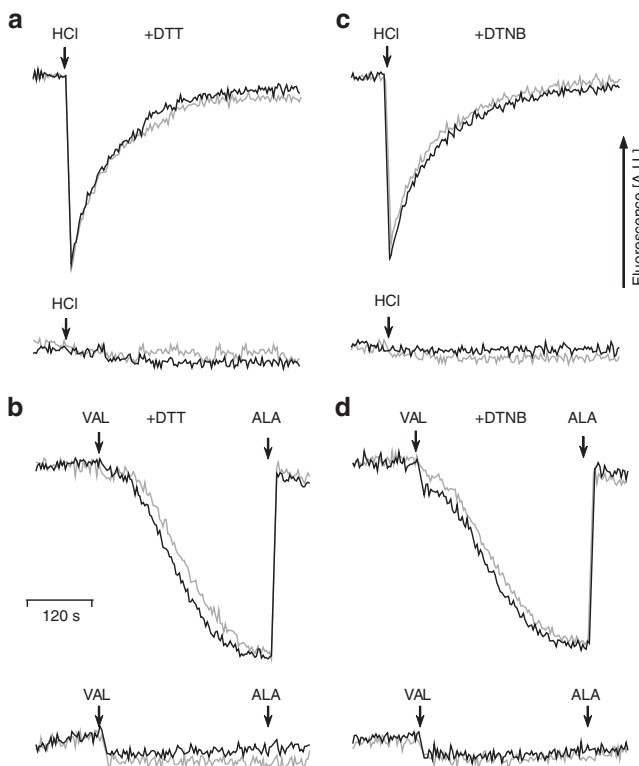

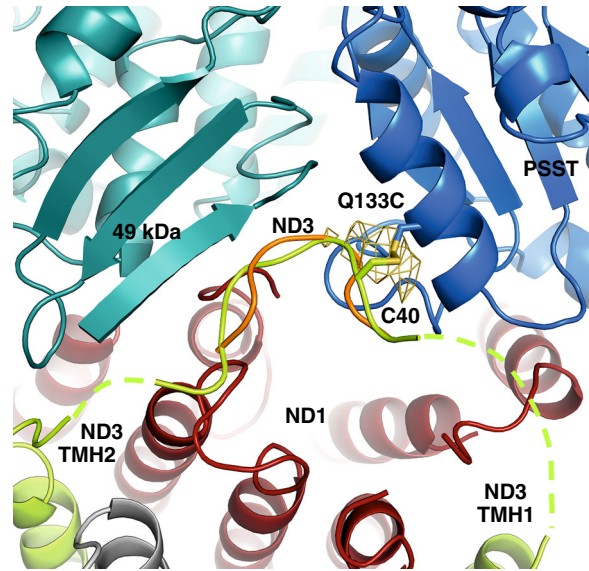

**Fig. 5** Formation of the disulfide cross-link immobilizes loop TMH1-2$^{ND3}$ without displacing it. Section of the crystal structure of complex I from mutant Q133C$^{PSST}$ showing the interface between membrane and peripheral arms in cartoon representation. The ND3 loop connecting TMH1 and TMH2 (light green) protrudes into the cavity formed by subunits PSST (blue), 49-kDa (blue-green), and ND1 (red). Residue C40$^{ND3}$ forms a disulfide bond with C133$^{PSST}$ (stick representation). This was confirmed by a strong peak in the sulfur anomalous difference Fourier electron density map (yellow mesh) which was absent when analyzing wild-type complex I crystals. Loop TMH1-2$^{ND3}$ from the superimposed structure of wild-type complex I [2] is shown in orange

**Fig. 4** The disulfide cross-link in mutant Q133C$^{PSST}$ does not change proton permeability. Passive proton (H$^+$) uptake was monitored as 9-amino-6-chloro-2-methoxyacridine (ACMA) fluorescence quench in proteoliposomes containing either parental (black lines) or mutant (gray lines) complex I treated with 5 mM dithiothreitol (DTT) (**a**, **b**) or 0.1 mM 5,5′-dithiobis-2-nitrobenzoic acid (DTNB) (**c**, **d**). In **a**, **c** 10 mM HCl were added to establish a transitory pH gradient. Proteoliposomes (5 μg ml$^{-1}$) were incubated for 5 min at room temperature in a reaction mixture containing 20 mM Mops pH 7.2 (Tris), 80 mM KCl and either 5 mM DTT or 0.1 mM DTNB. 0.5 μM ACMA and 0.5 μM valinomycin (VAL) were added immediately before recording the baseline. In **b**, **d** a K$^+$ gradient was used to drive complex I independent H$^+$ uptake. Proteoliposomes ([K$^+$]$_i$ = 80 mM) were incubated in 20 mM Mops pH 7.2 (Tris), 0.08 mM KCl, 160 mM sucrose and ΔΨ generation was induced by the addition of 0.5 μM valinomycin (VAL). A slow compensatory H$^+$ uptake was observed. Finally, 0.05 mg ml$^{-1}$ alamethicin (ALA) was added to permeabilize the proteoliposomes and collapse the ion gradients. For the bottom traces shown in all panels, 5 μM carbonylcyanide $m$-chlorophenylhydrazine (CCCP) were added before the assay to dissipate the proton gradient. Note that neither complex I substrates nor inhibitors were used in these assays. Representative traces of one out of five technical replicates are shown

machinery from the redox chemistry of ubiquinone and its control by the A/D transition in itself provides an invaluable tool to elucidate the mechanism and regulation of mitochondrial complex I.

## Methods
**Chemicals**. All chemicals were purchased from Sigma-Aldrich® unless stated otherwise.

**Strains, growth conditions, and site-directed mutagenesis**. All cloning procedures were performed as described earlier[25] and using enzymes and reagents from New England BioLabs® Inc. unless stated otherwise. Briefly, Q133C$^{PSST}$ was constructed by inverse PCR mutagenesis of pUB4/*NUKM*[25] in *Escherichia coli*. The desired point mutation was generated using appropriate primer pairs (Supplementary Table 2). The forward primer (NUKM_Q133Cf) had a mismatch codon at

the 5′ end leading to the point mutation. The reverse primer (NUKM_Q133r) ended just after the mismatch codon and contained no mismatch. Primers were phosphorylated using the T4 polynucleotide kinase kit prior to PCR. The PCR reaction mixture contained plasmid DNA harboring the *NUKM* gene as template (~10 ng), both primers (0.5 μM each), dNTPs (120 μM), Phusion HF buffer and Phusion DNA polymerase (2 units) in a final volume of 50 μl. The reaction was started by adding the Phusion DNA polymerase (manual hot start, 98 °C). PCR products were incubated with DpnI overnight at 37 °C to get rid of the original DNA template. Digested products were loaded onto a 1% agarose gel stained with ethidium bromide and separated at 120 V for 1 h. Linear PCR products (~10.3 kb) were cut under weak UV light and extracted from the gel spots using the Nucleospin® Gel and PCR clean-up kit from Macherey & Nagel. DNA concentration was measured using a NanoDrop 1000 spectrophotometer (Thermo Fisher Scientific). Cleaned products (100 ng DNA) were circularized by blunt-end ligation using the Fast-Link$^{TM}$ DNA ligation kit from Epicentre® following the manufacturer's instructions. Then, chemical-competent *E. coli* DH5α cells were transformed with one third of the ligation reaction volume (5 μl) by heat shock (30 s, 42 °C), incubated with SOC medium (0.5% yeast extract, 1% bacto-peptone, 1% NaCl, 10 mM MgCl$_2$, 20 mM dextrose) for 1 h at 37 °C under vigorous shaking and spread on selective LB (0.5% yeast extract, 1% bacto-peptone, 1% NaCl) + 100 μg ml$^{-1}$ ampicillin agar plates overnight at 37 °C. The next day, ten resistant clones were picked and grown individually in 3 ml LB + 100 μg ml$^{-1}$ ampicillin liquid medium for 16 h at 37 °C with vigorous shaking. Plasmid DNA was isolated using the QuickLyse Miniprep kit from QIAGEN® following the manufacturer's instructions. Purified plasmids were further digested with EcoRI and DNA fragments were separated in an agarose gel as above described. Positive clones were taken as the ones showing a two bands pattern at ~8 kb (empty pUB4 vector) and ~2.3 kb (*NUKM* gene). Before transformation into yeast cells, the entire *NUKM* open reading frame and the correctness of the point mutation were verified by DNA sequencing with primers NUKM-seq-1fw and NUKM-seq-2fw (Supplementary Table 2).

*Y. lipolytica* strain *nukm*Δ (*nukm::LEU2, nugm-Htg2, ndh2i, ura3⁻, lys11⁻*)[26] lacking the entire nuclear reading frame encoding subunit PSST was transformed with replicative plasmid pUB4 containing either wild-type or site-directed mutant copies of the *NUKM* gene under the control of its endogenous promoter. Transformation of *Y. lipolytica* was performed according to the one-step protocol described by Chen et al.[27]. Briefly, 500 μl of an overnight YPD (1% yeast extract, 2% bacto-peptone, and 2% dextrose) culture of *nukm*Δ cells were spun down, washed with deionized water and resuspended in 100 μl one-step buffer (45%(w/v) PEG 4000, 0.1 M lithium acetate, 0.1 M DTT, 0.25 mg ml$^{-1}$ ssDNA). Then, 100 ng

of plasmid DNA were added for each transformation mixture. The transformation cocktail was thoroughly vortexed and incubated at 39 °C for 60 min. The mixture was spread on a selective YPD + 50 µg ml$^{-1}$ Hygromycin B agar plate and incubated for 48–72 h at 28 °C. Five resistant clones were singularized and grown individually in baffled flasks containing 30 ml YPD + 50 µg ml$^{-1}$ Hygromycin B liquid medium for 24 h at 28 °C with vigorous shaking. The next day, pre-cultures were transferred into baffled flasks containing 450 ml YPD + 50 µg ml$^{-1}$ Hygromycin B liquid medium and the growth continued for 16 h under the same conditions. Yeast cells were harvested and used for mitochondrial membranes isolation. The Q133C$^{PSST}$ mutation was verified at the protein level by MS following separation by Blue Native-PAGE and in-gel digestion of the complex I band with trypsin. Haploid strain nukmΔ[26] complemented with pUB4 carrying the wild-type NUKM gene or (for experiments with purified complex I) haploid strain GB20 (mus51Δ, nugm-Htg2, ndh2i, lys11−, leu2−, ura3−, MatB) were used as controls and denoted parental throughout.

For crystallization of complex I we deleted the gene encoding subunit ST1 known to be present sub-stoichiometrically in the purified enzyme[28] from Y. lipolytica strain GB20 following the gene deletion and marker rescue strategy of Fickers et al.[29]. Specifically, for cloning of the flanking regions of the ST1 gene and introducing the Sce I sites for insertion of the loxR-URA3-loxP disruption cassette we used primer pairs ST1_P1/ST1_P2 and ST1_T1/ST1_T2, respectively (Supplementary Table 2). The NUKM gene was deleted from the haploid st1Δ strain (mus51Δ, st1Δ, nugm-Htg2, ndh2i, lys11−, leu2−, ura3−; denoted wild type in this study) by homologous recombination following the strategy described in Ahlers et al.[30] to obtain the double deletion strain nukmΔst1Δ (mus51Δ, nukm::LEU2, st1Δ, nugm-Htg2, ndh2i, ura3−, lys11−). Briefly, genomic DNA from strain nukmΔ was isolated and used as template to amplify the nukm::LEU2 allele (~2.5 kb) by PCR using primers nukmleu2_fw2 and nukmleu2_rv2 (Supplementary Table 2). The correct PCR product was extracted from the gel and cleaned as above described. Then, 500 µl overnight YPD culture of st1Δ cells were spun down, washed with deionized water, and resuspended in 100 µl one-step buffer. Yeast cells were further transformed with 100–200 ng of linear DNA and spread on SD (2% dextrose, 0.67% YNB w/o aminoacids, 50 mM phosphate buffer, pH 6.0) agar plates supplemented with 0.03 mg ml$^{-1}$ L-lysine and 0.02 mg ml$^{-1}$ uridine and incubated for 72–96 h at 28 °C. Fifteen LEU2 clones were screened for the correct homologous recombination at the NUKM gene locus by PCR using primers nukm5UTR_fw and leu2inv_rv (Supplementary Table 2) and using genomic DNA as template. Moreover, mitochondrial membranes from four putative nukmΔst1Δ clones were isolated in order to confirm the absence of assembled complex I by BN PAGE and activity measurements (see below). The correct double deletion strain was finally complemented with shuttle vector pUB4 carrying the mutated NUKM gene (Q133C$^{PSST}$).

**Preparation of mitochondrial membranes.** Small-scale isolation of unsealed mitochondrial membranes from parental and mutant strains was carried out essentially as described in Kerscher et al.[31] with minor modifications. Briefly, freshly harvested cells (~10 g wet weight) were resuspended in mitobuffer (600 mM sucrose, 20 mM K$^+$-Mops pH 7.2, 1 mM EDTA, 2 mM phenylmethylsulfonyl fluoride (PMSF)) and broken by vortexing (10 × 1 min with 1 min resting intervals on ice) in the presence of 10 g glass beads (0.45 mm). Glass beads, unbroken cells and nuclei were removed by centrifugation at 3300 × g (45 min, 4 °C). From supernatants, mitochondrial membranes were sedimented by centrifugation at 40,000 × g (120 min, 4 °C). Finally, the pellets were resuspended in 500 µl mito-buffer, homogenized and shock frozen as aliquots in liquid nitrogen and stored at −80 °C. Aliquots were thawed on ice before use. Protein concentration was determined using the DC protein assay kit (Bio-Rad).

For purification of complex I, Y. lipolytica cells were grown overnight at 28 °C in a 10 l fermenter (Biostat C, Sartorius) in YPD. After 16–20 h, cells were harvested by centrifugation for 10 min at 5000 × g and 4 °C. For large-scale mitochondrial membranes preparation, cells (200 g wet weight) were resuspended in 400 ml buffer A (600 mM sucrose, 1 mM EDTA, 20 mM Na$^+$/Mops, pH 7.2) in the presence of 1.5 mM PMSF. Cells were broken using a cooled Cell-Desintegrator-C, (Bernd Euler Biotechnologie) and 80 ml of 0.5 mm glass beads for 2 h. Cells debris was sedimented by centrifugation (30 min, 4400 × g, 4 °C) and mitochondrial membranes were obtained by ultracentrifugation (90 min, 167,000 × g, 4 °C). The pellet was resuspended in buffer A without EDTA and centrifuged again (60 min, 174,000 × g, 4 °C). Sedimented membranes were homogenized in buffer B (600 mM sucrose, 50 mM NaCl, 20 mM Na-Borate, 20 mM Na$^+$/Mops, pH 7.2, supplemented with 1.5 mM PMSF immediately prior to use), shock frozen in liquid nitrogen and stored at −80 °C.

**Purification of complex I.** Purification of n-dodecyl-β-D-maltoside (DDM) solubilized complex I from control and mutant strains was performed by Ni-affinity chromatography and gel filtration as described[32] with some modifications[33]. Briefly, mitochondrial membranes were thawed and adjusted to a concentration of 18 mg ml$^{-1}$ protein with 50 mM NaCl, 20 mM Na-Borate; pH 7.2, 1.5 mM PMSF. A total of 3400–4000 mg mitochondrial protein was typically used. DDM was added dropwise from a 20% (w/v) stock solution to a final detergent to protein ratio of 1 g:1 g. The solution was stirred on ice for 10 min and then centrifuged at 147,000 × g for 1 h. The supernatant was adjusted to 400 mM NaCl,

55 mM imidazole, 0.8 mM MgCl$_2$ and pH 7.3–7.4 prior to loading onto a 50 ml Ni-NTA Sepharose column (Bio-Rad) equilibrated with 55 mM imidazole, 400 mM NaCl, 0.025% DDM, and 20 mM Na-phosphate, pH 7.2. The column was washed with 150 ml of the same buffer and complex I was eluted with 200 ml of 140 mM imidazole, 400 mM NaCl, 0.025% DDM, and 20 mM Na-phosphate, pH 7.2. Complex I content of fractions was evaluated by measuring the electron transfer activity from NADH to the non-physiological electron acceptor hexaammineruthenium(III)-chloride (HAR). NADH oxidation ($\varepsilon_{340-400\ nm} = 6.22\ \text{mM}^{-1}\ \text{cm}^{-1}$) was measured in a reaction mix containing 20 mM Na/Hepes, pH 8.0, 250 mM sucrose, 2 mM NaN$_3$, 0.2 mM EDTA, 0.2 mM NADH, and 2 mM HAR in a Shimadzu UV-2450 sprectrophotometer. Peak fractions were combined and concentrated using Centriped centrifugal filter devices (Millipore®). For size exclusion chromatography, the concentrated pool was applied to a TSK gel G4000SW column (TosoH Bioscience). For equilibration and elution was performed with 100 mM NaCl, 1 mM EDTA, 20 mM Tris/HCl, pH 7.2 and 0.025% DDM. Chromatography was performed on an Äkta purifier chromatography system (GE Healthcare). Peak fractions were pooled, concentrated using spin devices, (Vivaspin, 100.000 MWCO, Sartorius) and aliquots were shock frozen and stored in liquid nitrogen.

For protein crystallization, complex I was purified essentially as described above but with some modifications[34]. During His-tag affinity chromatography the detergent was changed from 0.025% DDM to 0.015% of the polyoxyethylene detergent C$_{12}$E$_9$ (Thesit) and the column was washed with only 115 ml of the buffer containing Thesit instead of DDM.

**Gel electrophoresis.** For Blue-native (BN) PAGE, mitochondrial membranes (200 µg protein per lane) were solubilized with 2 g DDM per g protein in 50 mM imidazole-HCl pH 7.0, 500 mM 6-aminohexanoic acid, 1 mM EDTA, and centrifuged at 22,000 × g for 20 min at 4 °C. Supernatants were supplemented with Coomassie brilliant blue G-250 (Serva G) and protein complexes were separated in 4–16% acrylamide gradient gels as detailed in Wittig et al.[35] for about 3 h at 30 mA, 400 V, 10 W in a cold room (4–6 °C). Complex I in-gel staining was done according to Nübel et al.[36] by incubating the gel in 25 mM Tris/HCl, pH 7.4 supplemented with 2.5 mg ml$^{-1}$ nitrotetrazolium blue (NTB) and 1 mM NADH. After 30 min, the reaction was stopped by incubating the gel in 50% methanol, 10% acetic acid. Gels were further washed and maintained in deionized water indefinitely.

Where indicated, purified complex I (30 µg per lane) was diluted to 50 µl with 20 mM K$^+$-Mops pH 7.2, 80 mM KCl and treated with either 5 mM DTT or 0.1 mM 5,5′-dithiobis-(2-nitrobenzoic acid) (DTNB) for 5 min at room temperature prior to the addition of 20 µl of non-reducing sample buffer B (12% (w/v) SDS, 30% (w/v) glycerol, 0.05% Coomassie blue G-250, 150 mM Tris-HCl pH 7.0)[37]. In order to analyze the subunit composition of complex I, protein samples were loaded onto a 16% polyacrylamide Tricine-SDS-gel and separated under non-reducing conditions overnight at 120 V, room temperature. The gels were Coomassie-stained according to Schägger[37]. In short, gels were fixed in 50% methanol, 10% acetic acid, 100 mM ammonium acetate for 1 h, stained with 0.025% Coomassie dye in 10% acetic acid for 2 h, destained twice in 10% acetic acid (1 h each) and kept in deionized water for further usage.

For 2D Tricine-SDS-PAGE (dSDS-PAGE), the first dimension was performed under non-reducing conditions using 10% polyacrylamide gels. For the second-dimension, whole complex I lanes were excised, Coomassie-stained and incubated in 50 mM Tris-HCl pH 8.0, 1%(w/v) SDS, 1%(v/v) 2-mercaptoethanol for 30 min at room temperature to reduce disulfide bonds. Then the gel strips were placed on top of 16% polyacrylamide gels and the proteins were now separated under reducing conditions. After electrophoresis gels were silver-stained according to Schägger[37]. Briefly, gels were fixed as above described, washed twice with deionized water (1 h each), sensitized with 0.005% sodium thiosulfate for 1 h, incubated with 0.1% silver nitrate for 1 h, washed with deionized water for a few seconds and incubated with 0.036% formaldehyde, 2% sodium carbonate until the protein bands/spots were visible (1–2 min). In order to stop the staining, gels were incubated in 50 mM EDTA disodium salt, pH 4.6 for 5 min and later washed and maintained in deionized water indefinitely.

**Mass spectrometry.** Proteins were identified by MS after tryptic in-gel digestion essentially following the protocol described in Heide et al.[38] with slight modifications. In short, protein-containing gel slices were cut out and diced in smaller pieces before they were transferred to a 96-well filter flate (Millipore®, MABVN1250) adapted manually to a 96-well plate (MaxiSorp™ Nunc) as waste collector. Gel pieces were incubated several times with 50% methanol, 50 mM ammonium hydrogen carbonate (AHC) under gentle agitation until they were destained completely. Excess solution was removed by centrifugation (2500 × g, 20 s). In the next step, gel pieces were incubated with 120 µl of 10 mM DTT for 60 min. After removal of excess solution (2500 × g, 20 s), 120 µl of 30 mM chloroacetamide was added to each well and removed after 45 min. After a washing step with 50% methanol, 50 mM AHC, gel pieces were dried at room temperature for 45 min. The dried gel pieces were swollen by adding 20 µl of 5 ng µl$^{-1}$ trypsin (sequencing grade, Promega®), 50 mM AHC, 1 mM CaCl$_2$ per well for 30 min at 4 °C. Then, 50 µl of 50 mM AHC was added to cover the gel pieces, followed by an overnight incubation at 37 °C to digest the proteins. The peptide-containing supernatants were collected by centrifugation (2500 × g, 30 s) into a 96-well PCR

plate (Axygen®). The gel pieces in the filter plate were washed once with 50 µl of 30% acetonitrile, 3% formic acid for 20 min to elute the remaining peptides. The combined eluates were dried in a SpeedVac Concentrator Plus (Eppendorf). Prior to MS, peptides were resuspended in 20 µl of 5% acetonitrile, 0.5% formic acid.

Tryptic peptides were separated by reverse-phase liquid chromatography and analyzed by tandem MS in a Q-Exactive Orbitrap Mass Spectrometer equipped with an Easy nLC1000 nano-flow ultra-high-pressure liquid chromatography system (Thermo Fisher Scientific). Peptides were separated using a 100 µm ID × 15 cm length PicoTip[TM] EMITTER column (new objective) filled with ReproSil-Pur C18-AQ reverse-phase beads of 3 µm particle size and 120 A˚ pore size (Dr. Maisch GmbH, Germany) using linear gradients of 5–35% acetonitrile, 0.1% formic acid (30 min) at a flow rate of 300 nl min$^{-1}$, followed by 35–80% acetonitrile, 0.1% formic acid (5 min) at 600 nl min$^{-1}$ and a final column wash with 80% acetonitrile (5 min) at 600 nl min$^{-1}$. The mass spectrometer operated in positive ion mode switching automatically between MS and data-dependent MS2 of the top 20 most abundant precursor ions. Full-scan MS mode (400–1400 $m/z$) was set at a resolution of 70,000 $m/\Delta m$ with an automatic gain control target of $1 \times 10^6$ ions and a maximum injection time of 20 ms. Selected ions for MS/MS were analyzed using the following parameters: resolution 17,500 $m/\Delta m$, automatic gain control target $1 \times 10^5$; maximum injection time 50 ms; precursor isolation window 4.0 Th. Only precursor ions of charge $z = 2$ and $z = 3$ were selected for collision-induced dissociation. Normalized collision energy was set to 30% at a dynamic exclusion window of 60 s. A lock mass ion ($m/z = 445.12$) was used for internal calibration[39].

MS raw data files were analyzed using the MaxQuant software (v1.5.0.25) applying the settings detailed in Huynen et al.,[40] except that searching was done against an in-house compiled version of the *Y. lipolytica* protein database from NCBI including annotations of all known complex I subunits as well as the sequences of the pig trypsin and known contaminants, such as human keratins. In addition to the original sequence of subunit NUKM/PSST, an entry for this protein, in which the original glutamine 133 was changed to cysteine, was added manually. The limit for the false discovery rate determined by target-decoy database search was set to 0.01. Database searches were done with 20 ppm and 0.5 Da mass tolerances for precursor and fragmented ions, respectively. Trypsin (two missed cleavages allowed) was selected as the protease. Dynamic modifications included N-terminal acetylation and oxidation of methionine. Cystein carbamidomethylation was set as a fixed modification. Intensity-Based Absolute Quantification of proteins (iBAQ)-values were normalized between samples by dividing them by the sum of iBAQ-values from complex I subunits 51-kDa, 49-kDa and ND5, which were used as loading controls.

**Activity measurements**. Complex I activities were determined by measuring the initial oxidation rates of deamino-NADH (dNADH) using a SPECTRAmax PLUS$^{384}$ plate reader spectrophotometer (Molecular Devices) at 360 nm ($\varepsilon_{360nm}$ = 4.46 mM$^{-1}$ cm$^{-1}$). This wavelength was chosen as isosbestic point of the DTNB/ 2-nitro-5-thiobenzoate (TNB) redox spectra to avoid optical inference from these compounds. dNADH was used in assays with mitochondrial membranes since it is a specific substrate for complex I and cannot be oxidized by alternative NADH dehydrogenases[41]. Mitochondrial membranes were diluted to 0.5 mg protein ml$^{-1}$ in pre-incubation buffer (50 mM Tris-HCl pH 7.2, 80 mM KCl, 0.2 mM EDTA) and kept on ice. To measure dNADH:hexaammineruthenium(III)-chloride (HAR) oxidoreductase activity, aliquots of 12.5 µl membranes were added into 12.5 µl pre-incubation buffer supplemented with 5 mM DTT or 0.1 mM DTNB and incubated in the multi-well plate (Maxi-Sorp[TM] Nunc) for 5 min at room temperature. The reaction was started by the addition of 225 µl HAR activity buffer (50 mM Tris-HCl pH 8.5, 80 mM KCl, 0.2 mM EDTA, 1.1 mM NaCN, 220 µM dNADH, and 2.2 mM HAR) and monitored for 3 min at 25 °C. Both dNADH and HAR were added to the assay buffer immediately before starting the reaction. The final concentration of membranes was 25 µg protein ml$^{-1}$. To measure dNADH:decylubiquinone (DBQ) oxidoreductase activity, aliquots of 25 µl membranes were incubated with either 5 mM DTT or 0.1 mM DTNB for 5 min at room temperature. Note that irreversible and complete inactivation of parental complex I was observed after prolonged incubation (>30 min) with DTNB, since the reagent slowly reacted with free cysteine-thiols. One minute before finishing the incubations, 1 mM decylubiquinone was added to the samples. Then, the reaction was started by the addition of 225 µl DBQ activity buffer (50 mM Tris-HCl pH 8.5, 80 mM KCl, 0.2 mM EDTA, 1.1 mM NaCN, and 110 µM dNADH) and monitored for 3 min at 25 °C. The final protein concentration of membranes was 50 µg protein ml$^{-1}$. In both HAR and DBQ activity assays, the final volume of the reaction was 250 µl; path length: 0.6 cm. Inhibitor-sensitive dNADH:DBQ activities were calculated by subtracting the residual rate in the presence of 10 µM 2-*n*-decyl-quinazolin-4-yl-amine (DQA). To allow comparison between different preparations, all activities were normalized to their respective dNADH:HAR activities. IC$_{50}$ values and kinetic parameters (apparent $K_m$ and $V_{max}$) were determined as described[26]. For determination of kinetic parameters, data were fitted to the standard Michaelis-Menten equation using GraphPad Prism (v6.01). IC$_{50}$ was defined as the concentration of inhibitor required to decrease the inhibitor-sensitive complex I activity by 50%. Activities in the absence of either DQA or rotenone corrected for residual unspecific dNADH:DBQ oxidoreductase activity in strain *nukm*Δ were taken as 100%.

**EPR spectroscopy**. Low temperature cw-EPR spectra were obtained using a Bruker ESP 300E spectrometer equipped with a liquid helium continuous flow cryostat, ESR 900 (Oxford Instruments). Samples of purified complex I were diluted to 5 mg protein ml$^{-1}$ in 20 mM Hepes pH 7.5, 100 mM NaCl, incubated with either 5 mM DTT or 0.1 mM DTNB for 5 min on ice and then mixed with 2 mM NADH in the EPR tube and frozen in liquid nitrogen after 30 s reaction time. Spectra were recorded as described earlier[26] using instrument settings as listed in the figure legend.

**Proton pumping and activity of reconstituted complex I**. Purified complex I (200 µg) from either GB20 or mutant strain Q133C$^{PSST}$ was reconstituted into proteoliposomes at a protein-to-lipid ratio of 1:50 (w/w) following the protocol described in Dröse et al.[22,42] with one important modification. During detergent removal, the four incubations with BioBeads (Bio-Rad) were shortened to 30 min each. Prior to reconstitution of the enzyme, 10 mg ml$^{-1}$ asolectin from soybean were solubilized in 1.6% Octyl β-D-glucopyranoside, 20 mM K$^+$-Mops pH 7.2, 80 mM KCl. After detergent removal, proteoliposomes were collected by centrifugation at 100,000 × g (1 h, 4 °C) and dissolved carefully in 200 µl of 20 mM K$^+$-Mops pH 7.2, 80 mM KCl. Typically, proteoliposomes were prepared in parallel and analyzed on the same day. After reconstitution, protein concentration was determined using the DC protein assay kit (Bio-Rad) and the orientation of complex I in the membrane was evaluated by measuring NADH:HAR oxidoreductase activities before and after adding 0.05% (w/v) DDM to make the membranes permeable for the substrates. The fraction of peripheral arm facing outside was found to be consistently 65 ± 5% for both parental and mutant complex I. Proton pumping was monitored by ACMA fluorescence quenching essentially as described in Dröse et al.[22] except that the protocol was adapted to fluorescence measurement in 96-well black microplates (Flat bottom, non binding, Nunc). Fluorescence changes were monitored at 25 °C in a Spectramax M2e plate reader (Molecular Devices). Settings: $\lambda_{ex} = 430$ nm, $\lambda_{em} = 475$ nm, top read mode, auto cutoff, PMT medium. Proteoliposomes (5 µg protein) were diluted in 100 µl of assay buffer (20 mM K$^+$-Mops pH 7.2, 80 mM KCl, 0.5 µM valinomycin) and treated with either 5 mM DTT or 0.1 mM DTNB for 5 min at room temperature. Then, 20 µl aliquots from each sample were diluted to 200 µl with assay buffer yielding a final concentration of proteoliposomes 5 µg protein ml$^{-1}$. After starting the measurement 0.5 µM ACMA, 70 µM DBQ, 100 µM NADH, and 5 µM carbonyl cyanide *m*-chlorophenyl hydrazine (CCCP) were added subsequently after 60, 120, 180 and 360 s, respectively. NADH:DBQ oxidoreductase activities of the reconstituted enzyme in the presence and absence of 5 µM CCCP were determined at a final concentration of 5 µg protein ml$^{-1}$. The same protocol as for mitochondrial membranes was used, except that samples were diluted in 20 mM K$^+$-Mops pH 7.2, 80 mM KCl and that 100 µM NADH and 70 µM DBQ were added as substrates.

**Crystallography**. Crystals of complex I from wild-type and mutant Q133C$^{PSST}$ were grown essentially as described before[2]. Briefly, purified complex I from wild type and mutant was crystallized using the hanging-drop vapor diffusion method in 24-well plates (Crystalgen) at a protein concentration of 35 mg/ml at 19.5 °C; 0.2% Cymal 4 (Anatrace) and 1 mM NADPH were used as additives. Up to 1.75 mM phosphatidylcholine was added to the protein solution to achieve growth of large (>300 µm) single crystals. The protein solution was mixed in a 1:1 ratio with 7% PEG 3350, 12% glycerol, 40 mM Ca-acetate, pH 7.3. After 15 min at room temperature the sample was centrifuged at 16,000 × g for 5 min at 18 °C and the clear supernatant was pipetted on siliconized cover slides (Jena Bioscience) and placed over a series of well solutions containing 6% PEG 3350, 40 mM calcium-acetate, 12% glycerol with incremental changes from pH 6.2 to pH 7. After about one week, crystals were cryo-protected by transfer into 20% PEG 3350, 40 mM calcium-acetate, 30% glycerol pH 6.3, and frozen in liquid propane.

X-ray diffraction data were collected at 12.398 keV on beamline PXI (Swiss Light Source, Paul Scherrer Institut, Villigen, Switzerland). The beamline is equipped with an Eiger 16 M detector. In order to locate precisely the sulfur atoms in an anomalous Fourier electron density map, highly redundant data were collected using X-rays at 5.975 keV and 5.2 keV on beamline P13 operated by EMBL Hamburg at the PETRA III storage ring (DESY, Hamburg, Germany) using measurement conditions close to those used for native-SAD data collection[43]. All diffraction experiments were performed at 100 K.

Diffraction data were processed with XDS or Autoproc[44,45] and scaled with XSCALE or with Aimless[46]. Similarly to the wild-type structure[2], the diffraction data are strongly anisotropic and anisotropically truncated using STARANISO[47] and the anisotropy correction server[48]. The previously determined complex I structure from *Y. lipolytica* (PDB code: 4wz7)[2] was used as start model for the refinement of the Q133C$^{PSST}$ variant structure. The model was improved using iterative cycles of manual rebuilding with COOT[49] and refinement with BUSTER[50], resulting in a structure at the anisotropic resolution of 4.2 × 4.2 × 3.8 Å. In the final structure, the 14 central subunits mostly include side chains while accessory subunits were modelled as poly-alanine due to limited resolution. The structure was refined to final $R_{work}$ and $R_{free}$ values of 36.1% and 36.3% respectively with 87.03% and 9.82% of the residues in the most favored and allowed areas of the Ramachandran plot, while only 3.15% are in disallowed regions.

**Statistical analysis**. If not indicated otherwise, data were analyzed by two-way ANOVA with Bonferroni correction. Results are presented as mean value ± standard deviation. Statistical significance is indicated as follows: $*p < 0.05$, $**p < 0.01$, $***p < 0.001$, $****p < 0.0001$.

## Data availability

The X-ray structure of complex I from *Yarrowia lipolytica* mutant Q133C$^{PSST}$ as well as the structure factors have been deposited in the Protein Data Bank with accession code 6H8K [http://www.rcsb.org/structure/6H8K]. All other data supporting the findings of the current study are available from the corresponding author on reasonable request.

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

## Acknowledgements

This work was supported by grants of the Deutsche Forschungsgemeinschaft to U.B., V.Z., and C.H. (SPP1710-BR 1633/3-1, ZI552/4-1, CRC746), the Netherlands Organization for Scientific Research to U.B. (TOP 714.017.00 4) and of the Excellence Initiative of the German Federal and State Governments to C.H. (EXC 294 BIOSS) and U.B. and V.Z. (EXC 115 CEF). We are indebted to Jean M. Nicaud for providing the constructs for the Cre-lox system. We thank Dr. Heike Angerer and Dr. Katarzyna Kmita for valuable support.

## Author contributions

U.B. and A.C.O. designed the study and interpreted the results; A.C.O., E.G.Y., K.S., S.G.C., and K.Z. performed the experiments; C.W., V.Z., and E.G.Y. collected the X-ray data; C.W. and C.H. performed the crystallographic analysis; U.B. and A.C.O. wrote the paper with contributions from V.Z., C.W., and C.H.; All authors read, corrected and approved the final version.

## Additional information

**Competing interests:** The authors declare no competing interests.

