## [Peer Review File · Nature Communications]

Reviewers' comments:

Reviewer #1 (Remarks to the Author):

The manuscript by Brandt and coworkers describes a very interesting mutation in *Yarrowia* complex I that can be used to reversibly link a well-known cysteine (on a loop between two transmembrane helices in ND3) to subunit PSST, to restrict the movement of the loop, and thereby study its effects on catalysis. This is very neat. In the resulting complex it is concluded that electron transfer, which is little affected, is no longer coupled to proton translocation, and therefore that movement of the ND3 loop is required for energy conversion – a striking conclusion. However, I am not convinced about the section that presents results relating to the active/deactive transition.

Specific comments:

1. The authors claim their observations corroborate their earlier two-state stabilization charge mechanism (refs 5 and 6). Ref. 6 describes this mechanism as follows: "During turnover, these loops perform a coordinated rearrangement resulting in a shift of the ubiquinone binding site and movement of the cluster of negative charges in loop TMH5–6 ND1, which may trigger an electrostatic pulse toward the membrane arm. Stabilization of the anionic species in the site leads to transition from E state (left) to P state (right), driving a stroke of proton pumping. The idling enzyme can convert reversibly from the active A form into the deactive D form with a structure similar to the P state." However, all the authors have done here is shown that restricting the movement of one of these loops prevents energy conversion – a small part of the mechanism. They should therefore not claim anything more than being consistent with their proposed mechanism.
2. Movement in the three loops discussed (in ND3, ND1 and 49 kDa) has been assigned as the basis of the active/deactive transition in mammalian complex I by comparison of structural data on the two specific states (Blaza 2018, Agip 2018). These structures must be discussed – first, because the same three loops are involved, and second, because it is relevant to understand how the *Yarrowia* structure presented here (specifically the three loop conformations) compare to both of them. Based on the comparison, can it be confirmed (for example, because the densities for the loops in question unambiguously adopt different conformations) that the new *Yarrowia* structure does not correspond to an active-like state? If it does not, how strong is the correspondence with the known mammalian deactive state?
3. It appears that the ND3 loop in the original *Yarrowia* structure was poorly resolved. In order to clarify the extent of this, and to confirm confidence in it remaining in the same place in the new structure, please provide the densities for the stretch of sequence comprising TMH1-loop-TMH2 for both structures as extended data.
4. Forming the disulfide is concluded to 'disconnect' the redox reaction from the proton pumping machinery – since the former proceeds as usual while the latter apparently does not. Notably, the deactive form of mammalian complex I was reported previously (Roberts 2012) to display antiporter-like behavior – as a result of disconnection of the redox reaction from the proton pumping machinery. In this case the proton pumping subunits were active while the redox reaction was prevented. This earlier work must be discussed.

5. It is formally possible here that the variant enzyme is pumping protons but also leaking them back. This possibility could be excluded by showing that it is possible to produce and maintain a PMF in vesicles containing the cross-linked enzyme to the same extent as in WT vesicles (e.g. by creating an ion gradient and using it to drive proton uptake independent of complex I).
6. Figure 1 – is this isolated complex I? In b, what is relative abundance? What is it relative to and how was it calculated?
7. Are the two cysteines that become cross-linked exposed to solvent in the structure to allow access of DTNB or DTT? Please include a statement on the mechanism of cross-linking by DTNB. Can the formation of stable DTNB adducts be excluded?
8. In Table 1 and ED Table 1 please include values from untreated complexes. In Table 1 – what substrate does KM refer to? What substrate concentrations were the inhibitor curves determined at?
9. It cannot be concluded that there are no effects on the FeS clusters or ET activity from a single EPR spectrum – for example, a cluster reduction potential could have changed. The difference in the spectrum with DTT and DTNB looks like a difference in the N1b signal. Please include corresponding spectra at 40 K to clarify this.
10. I am not comfortable with the discussion of the effects of Mg^{2+} on catalysis described in Figure 2. Viewed objectively, these data could simply show Mg^{2+} inhibits catalysis independently of the presence of the cross link. Please show examples of assay traces that show how, for both *Yarrowia* isolated complex I and membranes, the observed lag phase expected from starting in the deactive state is prolonged in the presence of Mg^{2+} . As shown in ref. 15 Mg^{2+} do not prevent reactivation, they only slow it down - since the rate observed evolves over time, at what point were the observed rates measured? On this basis, plus the lack of structural comparisons and interpretations referring to the known deactive and active states, I do not currently agree that the data presented show that the cross-link has no effect on the D/A transition.
11. As the ACMA response is non-quantitative the comparison between Figure 3a and b is difficult – and it is further complicated by the ‘vertical’ drop in fluorescence in Figure 3a (which presents essentially as an infinite rate). It is possible that proton translocation in the mutant is severely affected – and therefore the on/off effect of forming the cross link is less clear than suggested. Perhaps by varying the conditions (such as concentrations of ACMA, permeant ions etc. - which should also be stated in the legend) the rate of response in Figure 3a could be brought on-scale for a more meaningful comparison?
12. What were the conditions that ‘favor formation of the disulfide’? Please include information on how complex I was pretreated by DTT and DTNB, including concentrations of the reagents (only membranes are mentioned on page 11). How were the concentrations set differently when used in pre-treatments or in the various assays to give matching effects?

Reviewer #2 (Remarks to the Author):

NCOMMS-18-18529-T

This very interesting manuscript describes the construction of a variant of complex I from the yeast *Yarrowia lipolytica* where a Cys residue is introduced into the PSST subunit at glutamine position 133. The introduced Q133C residue is then able to form a disulfide crosslink with Cys40 of the ND3 subunit following treatment with DTNB. The presence of the crosslink is convincingly shown by the data in Figure 1 and the x-ray crystal structure of the variant (Figure 4). Cys40 or

ND3 has been previously shown to be a residue particularly sensitive to modification by the sulfhydryl inhibitor NEM when complex I is locked into its Deactive form (D-form). It is quite interesting that formation of the crosslink does not appear to have any effect on catalytic activity of complex I (either HAR reductase activity or decylubiquinone activity), sensitivity to Q-site inhibitors, or apparently the A/D-transition. In fact, Fig. 2b suggests that the crosslink desensitizes the D-form of the enzyme to NEM inhibition. Importantly and the apparent key finding of the paper the crosslinked enzyme is no longer able to efficiently pump protons (Fig. 3). The authors interpret this based upon their previously proposed model where negatively charged ubiquinone intermediates at the Q-site are associated with movement of protein loops near the site of the crosslinked region between PSST and ND3 drive the pumping of protons through the P-module of complex I.

Specific comments:

1. Although the authors suggestion seems reasonable and can be interpreted exactly as they have done why couldn't the data from Figure 3d also be interpreted as the introduction of a proton leak? It seems the paper might be improved somewhat by at least indicating alternative interpretations of the data or explain to this reader why Fig. 3d could not alternatively been interpreted as a leak? Maybe this is why in extended data Table 1 addition of CCCP to the crosslinked Q133C variant indicates that the protein is uncoupled.
2. Although probably not essential data to add to this paper, interesting information if available, is whether or not the crosslinked enzyme (otherwise apparently fully active) is capable of reverse electron transport? Since RET requires a membrane potential presumably this could be looked at in submitochondrial particles prepared from the Q133C(psst) variant. Any comment regarding this would certainly be of interest to the reader.
3. If a reader is not fully familiar with the complex I field it may be difficult to understand the power stroke model invoked to drive the proton pumps as envisioned in Refs. 4 and 5. Possibly showing this model in the manuscript (or at least the extended data set) would help the reader by not requiring the visit to the previous publications.
4. Although a minor comment in the legend to Figure 4 the color codes used seem somewhat problematic for the non-native English speaker. Are...limon... andmarine.... acceptable indications of color? Although these are listed in the color palettes of some drawing programs it would just seem that ...light green... andblue.... would be preferred indications for the colors.

Reviewer #3 (Remarks to the Author):

This is an excellent and very interesting piece of work. In very well-controlled experiments it is demonstrated that fixing an interhelical loop domain by the use of a reversible sulfur bridge essentially decouples the proton pump function of mitochondrial complex I from the redox reaction. This is the first time such a phenotype is achieved with complex I, and is a very valuable tool for further detailed understanding of the proton pump mechanism.

In the description of the current status of knowledge (lines 80-90) it may be appropriate to mention (and cite) the origins of the proposal that it is the negative charge of ubiquinone intermediates that triggers the pumping mechanism via electrostatic and conformational changes. Likewise, it may strengthen the case to mention that computational studies have also indicated the importance of structural changes in the three loop domains considered by Zickermann et al. (Science 2015); (see Sharma et al. [2015] PNAS 112, 11571).

Finally, it would be of considerable interest if the authors would discuss at least to some extent what the atomistic consequences might be of their finding with regard to the proton pump mechanism. Tinkering with that mechanism can, in principle, lead either to a blockade of the redox activity, or (as is the case here) to "decoupling" of the mechanism. How that might occur as a result of "fixing" the interhelical loop would deserve at least some comment even if it were to be speculative.

Reviewer #1 (Remarks to the Author):

The manuscript by Brandt and coworkers describes a very interesting mutation in *Yarrowia* complex I that can be used to reversibly link a well-known cysteine (on a loop between two transmembrane helices in ND3) to subunit PSST, to restrict the movement of the loop, and thereby study its effects on catalysis. This is very neat. In the resulting complex it is concluded that electron transfer, which is little affected, is no longer coupled to proton translocation, and therefore that movement of the ND3 loop is required for energy conversion – a striking conclusion. However, I am not convinced about the section that presents results relating to the active/deactive transition.

Specific comments:

1. The authors claim their observations corroborate their earlier two-state stabilization charge mechanism (refs 5 and 6). Ref. 6 describes this mechanism as follows: “During turnover, these loops perform a coordinated rearrangement resulting in a shift of the ubiquinone binding site and movement of the cluster of negative charges in loop TMH5–6 ND1, which may trigger an electrostatic pulse toward the membrane arm. Stabilization of the anionic species in the site leads to transition from E state (left) to P state (right), driving a stroke of proton pumping. The idling enzyme can convert reversibly from the active A form into the deactive D form with a structure similar to the P state.” However, all the authors have done here is shown that restricting the movement of one of these loops prevents energy conversion – a small part of the mechanism. They should therefore not claim anything more than being consistent with their proposed mechanism.

We thank the reviewer for this remark, since we did not intend to imply that our results corroborate any predictions of our mechanistic hypothesis other than that movement of one of the loops is required to drive proton pumping. We have rewritten the Abstract and the summary statement at the end of the main text to make this very clear.

2. Movement in the three loops discussed (in ND3, ND1 and 49 kDa) has been assigned as the basis of the active/deactive transition in mammalian complex I by comparison of structural data on the two specific states (Blaza 2018, Agip 2018). These structures must be discussed – first, because the same three loops are involved, and second, because it is relevant to understand how the *Yarrowia* structure presented here (specifically the three loop conformations) compare to both of them. Based on the comparison, can it be confirmed (for example, because the densities for the loops in question unambiguously adopt different conformations) that the new *Yarrowia* structure does not correspond to an active-like state? If it does not, how strong is the correspondence with the known mammalian deactive state?

*Although the structural differences between the active and deactive form of mitochondrial complex I are not the central topic of this study, we expanded this aspect in the introduction and now discuss the recent cryoEM structures (Blaza et al 2018; Agip et al. 2018). Our previous X-ray structure of *Y. lipolytica* (Zickermann et al 2015) was clearly in the deactive form, but it has been argued by Blaza et al, that the critical loops were more ordered because a ubiquinone-site inhibitor was present in the crystals. We assume that this is the “active-like” state the reviewer has in mind. However, the new native structure was made in the absence of any inhibitor and it still shows the loop in the same position. This is also the case in a new structure of deactive *Y. lipolytica* complex I without inhibitor that we have now obtained by cryoEM (Parey et al., submitted).*

3. It appears that the ND3 loop in the original *Yarrowia* structure was poorly resolved. In order to clarify the extent of this, and to confirm confidence in it remaining in the same place in the new

structure, please provide the densities for the stretch of sequence comprising TMH1-loop-TMH2 for both structures as extended data.

As requested, we have added a stereo view of the density of the ND3 loop in the wild type enzyme to Extended Data Figure 6. It clearly shows that the loop is in the same position in both structures.

4. Forming the disulfide is concluded to 'disconnect' the redox reaction from the proton pumping machinery – since the former proceeds as usual while the latter apparently does not. Notably, the deactive form of mammalian complex I was reported previously (Roberts 2012) to display antiporter-like behavior – as a result of disconnection of the redox reaction from the proton pumping machinery. In this case the proton pumping subunits were active while the redox reaction was prevented. This earlier work must be discussed.

We now briefly discuss the finding of Roberts et al. in the context of the additional data provided in response to point 5 of reviewer #1 and point 1 of reviewer #2.

5. It is formally possible here that the variant enzyme is pumping protons but also leaking them back. This possibility could be excluded by showing that it is possible to produce and maintain a PMF in vesicles containing the cross-linked enzyme to the same extent as in WT vesicles (e.g. by creating an ion gradient and using it to drive proton uptake independent of complex I).

We thank the reviewer for this very helpful suggestion and have added new data showing in two ways that dislodging the proton pump does not increase proton permeability of the complex I proteoliposomes.

6. Figure 1 – is this isolated complex I? In b, what is relative abundance? What is it relative to and how was it calculated?

The Figure legend states that this experiment was performed with purified complex I. The legend was amended to better explain the origin of the data shown in panel b.

7. Are the two cysteines that become cross-linked exposed to solvent in the structure to allow access of DTNB or DTT? Please include a statement on the mechanism of cross-linking by DTNB. Can the formation of stable DTNB adducts be excluded?

*In the absence of substrates, complex I from *Y. lipolytica* quickly and completely converts into the deactive form, not requiring elevated temperature as the mammalian enzyme. Accessibility of C40^{ND3} for thiol agents is one of the hallmarks of the deactive forms and has been demonstrated also for the fungal complex. In fact, our data as such clearly demonstrate that both agents could access the two cysteines, since we could use them to efficiently open and close the disulfide bridge between them. In using DTNB to induce disulfide formation between two adjacent cysteines, we followed an established protocol (Ref. 16). These authors proposed that DTNB transiently binds to one of the cysteines thereby activating it and is then displaced by the second cysteine. As explained in the methods section, the DTNB adduct indeed accumulate during extended incubation with the reagent and then inhibit complex I like other thiol agents by modifying C40^{ND3} of the deactive enzyme. The mere fact that activity was not significantly reduced during the 5 min incubation time we used in our experiments thus in itself demonstrates that stable modification of C40^{ND3} could be neglected under the conditions used.*

8. In Table 1 and ED Table 1 please include values from untreated complexes. In Table 1 – what substrate does KM refer to? What substrate concentrations were the inhibitor curves determined at?

This information was added to the Tables as requested.

9. It cannot be concluded that there are no effects on the FeS clusters or ET activity from a single EPR spectrum – for example, a cluster reduction potential could have changed. The difference in the spectrum with DTT and DTNB looks like a difference in the N1b signal. Please include corresponding spectra at 40 K to clarify this.

No effect on the EPR signatures of the iron-sulfur clusters were observed also at 40K and 5K. These spectra are now also shown as Extended Data. Midpoint potentials were not determined, but a change of this parameter seems rather unlikely given that the mutation is located remotely from the clusters and that there is no indication for changes in their local environment and redox steady state kinetics are not affected. We therefore considered further exploring this aspect as beyond the scope of the current study. However, we carefully amended the text to express more precisely, what can be concluded from the EPR analysis.

10. I am not comfortable with the discussion of the effects of Mg²⁺ on catalysis described in Figure 2. Viewed objectively, these data could simply show Mg²⁺ inhibits catalysis independently of the presence of the cross link. Please show examples of assay traces that show how, for both *Yarrowia* isolated complex I and membranes, the observed lag phase expected from starting in the deactive state is prolonged in the presence of Mg²⁺. As shown in ref. 15 Mg²⁺ do not prevent reactivation, they only slow it down - since the rate observed evolves over time, at what point were the observed rates measured? On this basis, plus the lack of structural comparisons and interpretations referring to the known deactive and active states, I do not currently agree that the data presented show that the cross-link has no effect on the D/A transition.

*These are all valid concerns for mammalian complex I. However, the Mg²⁺ inhibition of the deactive form of complex I is much more efficient and under the conditions used stable for at least three to five minutes. In fact, reactivation is not just delayed by Mg²⁺ with *Y. lipolytica* complex I, i.e. dNADH oxidation rates did not speed up over the time for at least 10 min under the experimental conditions used. To make this clear, we have added original traces of experiments performed with parental and mutant complex I as extended data, which demonstrate that the assay used indeed monitors the A/D transition of the complex.*

11. As the ACMA response is non-quantitative the comparison between Figure 3a and b is difficult – and it is further complicated by the ‘vertical’ drop in fluorescence in Figure 3a (which presents essentially as an infinite rate). It is possible that proton translocation in the mutant is severely affected – and therefore the on/off effect of forming the cross link is less clear than suggested. Perhaps by varying the conditions (such as concentrations of ACMA, permeant ions etc. - which should also be stated in the legend) the rate of response in Figure 3a could be brought on-scale for a more meaningful comparison?

Indeed, it would be interesting to study the rates of proton pumping in a quantitative fashion, which as the reviewer indicates is not trivial using ACMA. Therefore, we consider a detailed quantitative assessment of the pump rates as beyond the scope of the study. We do think that the data as presented show very clearly, though only in a qualitative fashion, that formation of the disulfide bridge in the mutants shuts off the proton pumps of complex I. As requested concentrations are now given in the figure legend and not only in the Methods section.

12. What were the conditions that ‘favor formation of the disulfide’? Please include information on how complex I was pretreated by DTT and DTNB, including concentrations of the reagents (only membranes are mentioned on page 11). How were the concentrations set differently when used in pre-treatments or in the various assays to give matching effects?

This information had been given in the Methods section and was not repeated in the figure legends as the same concentrations of DTT and DTNB as well as incubation time was used throughout. However for clarity and as requested by the reviewer this information is now found in all legends.

Reviewer #2 (Remarks to the Author):

This very interesting manuscript describes the construction of a variant of complex I from the yeast *Yarrowia lipolytica* where a Cys residue is introduced into the PSST subunit at glutamine position 133. The introduced Q133C residue is then able to form a disulfide crosslink with Cys40 of the ND3 subunit following treatment with DTNB. The presence of the crosslink is convincingly shown by the data in Figure 1 and the x-ray crystal structure of the variant (Figure 4). Cys40 or ND3 has been previously shown to be a residue particularly sensitive to modification by the sulfhydryl inhibitor NEM when complex I is locked into its Deactive form (D-form). It is quite interesting that formation of the crosslink does not appear to have any effect on catalytic activity of complex I (either HAR reductase activity of decylubiquinone activity), sensitivity to Q-site inhibitors, or apparently the A/D-transition. In fact, Fig. 2b suggests that the crosslink desensitizes the D-form of the enzyme to NEM inhibition. Importantly and the apparent key finding of the paper the crosslinked enzyme is no longer to efficiently pump protons (Fig. 3). The authors interpret this based upon their previously proposed model where negatively charged ubiquinone intermediates at the Q-site are associated with movement of protein loops near the site of the crosslinked region between PSST and ND3 drive the pumping of protons through the P-module of complex I.

Specific comments:

1. Although the authors suggestion seems reasonable and can be interpreted exactly as they have done why couldn't the data from Figure 3d also be interpreted as the introduction of a proton leak? It seems the paper might be improved somewhat by at least indicating alternative interpretations of the data or explain to this reader why Fig. 3d could not alternatively been interpreted as a leak? Maybe this is why in extended data Table 1 addition of CCCP to the crosslinked Q133C variant indicates that the protein is uncoupled.

We thank the reviewer for this very helpful suggestion and have added new data showing in two ways that dislodging the proton pump does not increase proton permeability of the complex I proteoliposomes.

2. Although probably not essential data to add to this paper, interesting information if available, is whether or not the crosslinked enzyme (otherwise apparently fully active) is capable of reverse electron transport? Since RET requires a membrane potential presumably this could be looked at in submitochondrial particles prepared from the Q133C(psst) variant. Any comment regarding this would certainly be of interest to the reader.

*This is a great suggestion but unfortunately, we are still struggling to establish assays to measure reverse electron transport with *Y. lipolytica* complex I.*

3. If a reader is not fully familiar with the complex I field it may be difficult to understand the power stroke model invoked to drive the proton pumps as envisioned in Refs. 4 and 5. Possibly showing this model in the manuscript (or at least the extended data set) would help the reader by not requiring the visit to the previous publications.

As requested we have added a schematic overview of the functional modules and the catalytic cycle of our hypothetical mechanism as Extended Data Figure 1b

4. Although a minor comment in the legend to Figure 4 the color codes used seem somewhat problematic for the non-native English speaker. Are...limon... andmarine.... acceptable indications

of color? Although these are listed in the color palettes of some drawing programs it would just seem that ...light green... andblue.... would be preferred indications for the colors.

As requested more common color descriptions are now used.

Reviewer #3 (Remarks to the Author):

This is an excellent and very interesting piece of work. In very well-controlled experiments it is demonstrated that fixing an interhelical loop domain by the use of a reversible sulfur bridge essentially decouples the proton pump function of mitochondrial complex I from the redox reaction. This is the first time such a phenotype is achieved with complex I, and is a very valuable tool for further detailed understanding of the proton pump mechanism.

In the description of the current status of knowledge (lines 80-90) it may be appropriate to mention (and cite) the origins of the proposal that it is the negative charge of ubiquinone intermediates that triggers the pumping mechanism via electrostatic and conformational changes. Likewise, it may strengthen the case to mention that computational studies have also indicated the importance of structural changes in the three loop domains considered by Zickermann et al. (Science 2015);(see Sharma et al. [2015] PNAS 112, 11571).

We have added brief discussion on mechanistic ideas concerning the pump modules itself and reference to the recent modeling work to the introduction. However, we find it difficult to trace back the idea that negatively charged ubiquinone intermediates are critical for the pumping mechanism to a single person or publication, since different variations of this theme have been discussed by many people already for a very long time.

Finally, it would be of considerable interest if the authors would discuss at least to some extent what the atomistic consequences might be of their finding with regard to the proton pump mechanism. Tinkering with that mechanism can, in principle, lead either to a blockade of the redox activity, or (as is the case here) to "decoupling" of the mechanism. How that might occur as a result of "fixing" the interhelical loop would deserve at least some comment even if it were to be speculative.

To accommodate the reviewer's suggestion, we have added a brief discussion at the end of the manuscript addressing the ramifications and possible structural implications of our findings.

REVIEWERS' COMMENTS:

Reviewer #1 (Remarks to the Author):

In general the authors have addressed my comments adequately. However, the referencing in the first paragraph of the introduction is a travesty: "Recent structures of mitochondrial complex I....substantiated earlier studies (refs 7, 8) showing that FMN and a chain of seven iron-sulfur clusters....". Refs 7 and 8 did not add anything to the locations of the FMN and FeS clusters; Ref 8 described the incorrect location of one of the core subunits. The locations of the FMN and FeS clusters were revealed by Sazanov and coworkers in a seminal piece of work on the bacterial complex, that is not referred to here at all. This is unacceptable.

REVIEWERS' COMMENTS:

Reviewer #1 (Remarks to the Author):

In general the authors have addressed my comments adequately. However, the referencing in the first paragraph of the introduction is a travesty: "Recent structures of mitochondrial complex I...substantiated earlier studies (refs 7, 8) showing that FMN and a chain of seven iron-sulfur clusters...". Refs 7 and 8 did not add anything to the locations of the FMN and FeS clusters; Ref 8 described the incorrect location of one of the core subunits. The locations of the FMN and FeS clusters were revealed by Sazanov and coworkers in a seminal piece of work on the bacterial complex, that is not referred to here at all. This is unacceptable.

We did not intend to make the claims implicated by the reviewer and do not agree completely with the harsh criticism. However, we concede that the criticized sentence was somewhat misleading. We have added a reference also to the work on bacterial complex I by Sazanov and coworkers. We amended the text to accommodate the criticism and state clearer, why we cited Refs. 7 and 8.